# Padel Match Analysis: Notational and Time-Motion Analysis during Official Italian Sub-Elite Competitions

**DOI:** 10.3390/ijerph19148386

**Published:** 2022-07-08

**Authors:** Alexandru Nicolae Ungureanu, Corrado Lupo, Paolo Riccardo Brustio

**Affiliations:** 1Department of Medical Science, University of Turin, 10126 Turin, Italy; alexandru.ungureanu@unito.it; 2NeuroMuscular Function Research Group, School of Exercise & Sport Sciences, University of Turin, 10126 Turin, Italy; paoloriccardo.brustio@unito.it; 3Department of Neuroscience, Biomedicine and Movement, University of Verona, 37129 Verona, Italy; 4Department of Clinical and Biological Sciences, University of Turin, 10126 Turin, Italy

**Keywords:** notational analysis, time-motion analysis, training load, padel tennis, key performance indicators, racket sports

## Abstract

Although performance analysis in padel represents a useful process to gain references about players’ technical and tactical behavior, most of the research was conducted in elite compared to the sub-elite competitions. Therefore, this study aimed to describe sub-elite competitions in order to enhance scientific knowledge for sub-elite athletes and technical staff. 4287 shots were analyzed within five areas (time-motion analysis, shots characteristics, errors, serve and points won). Effective playing time and work-to-rest ratio were lower than in elite competitions, while strokes per minute and total match duration were in line with it. Shots were mainly forehand volleys performed under the head, while volleys and smashes were more likely to end with a point in comparison with ground or wall shots. However, sub-elite winning pairs performed fewer volleys than the losing side and fewer errors on volleys. One serve out of five ended in errors (almost half were net errors); fewer errors during serve return shots represented an advantage for the winning pairs. Finally, 65% of the points scored were caused by unforced errors of opponents. This knowledge should help technical staff design specific training programs for sub-elite padel players.

## 1. Introduction

Padel is a doubles racket sport that was born in 1969 in Acapulco (Mexico), and spread in Argentina and Spain to become very widespread across Europe in the present day [1]. Due to the great popularity of this sport across Europe, the major European stakeholders (i.e., Spanish, Finnish, British, Swiss, Polish, Danish, Portuguese, Austrian, Czech, Belgium, Deutscher, Nederlandse, Svenska and Estonian Federations) created the European Padel Association [2]. In particular, the Italian Olympic Committee recognized padel as a stand-alone discipline, and consequently, the participation of the athletes in national and international competitions. Moreover, the ongoing development of padel increased amateur and competitive level practitioners’ participation while decreasing tennis practitioners’ participation, especially in Spain [3].

Padel is a netball and racquet game similar to tennis, in terms of its scoring system, but with some changes, such as the underhand serve and the characteristics of the court [4]. Padel is played on a 20 m × 10 m (length × width) enclosed synthetic glass and metal court divided by a standard tennis net (0.88 m at the center strap and 0.92 m at the post) in the middle [4]. The back (3 m height × 10 m length) and the side walls (3 m × 2 m) end on another 2 m × 2 m wall, while the rest of the court consists of two metallic panels of equal dimensions (3 m × 2.59 m) and one gate (2 m × 0.82 m) for each half [4]. This setting allows the ball to bounce on lateral and back walls [4], and leads to longer rallies than other racket sports such as tennis or badminton [5]. However, substantial differences in rally durations exist according to age, competition level or type [5]. Performance in padel was widely investigated in elite competitions [6,7,8,9,10], and scientific research has arisen to better understand the characteristics of padel in terms of anthropometrics, biomechanics, epidemiology, physiological requirements and match analysis [10]. In this regard, the literature shows controversial results about time-motion (TMA) and notational analysis (NA). The effective playing time is around 56 min [11], and it varies between 38% [12] and 46% [13] of the total match duration, while ball in play and break time per rally are roughly 13 and 15 s, respectively, with a work-to-rest ratio (WRR) of 0.84 on average [13]. The main distribution of rallies duration is commonly between 3 to 6 s (23.2%), 6 to 9 s (29.3%), 9 to 12 s (19.6%) and 12 to 15 s (13.3%) [5]. Nevertheless, when comparing male and female athletes, controversial results exist. In particular, Lupo et al. [6] reported significant and considerable differences in the rallies’ duration between men and women (i.e., 12.6 s vs. 16.8 s), while Torres-Luque et al. [5] did not (i.e., 9.3 s vs. 9.7 s). Similarly, the number of strokes per rally was 9.3 and 9.5 for men and women, respectively, with no differences for gender [5]. On the contrary, Lupo et al. reported large differences between men and women, of 9.6 vs. 12.2, respectively. These divergences in literature can be due to the competition level, as performance is level-dependent in padel [14,15]. In fact, when comparing final or semi-final matches (such as Torres-Luque et al. [5]) to other high levels, even professional matches (such as Lupo et al. [6]), discrepancies may emerge, and this evidence stresses the need for comparing the same level of competition, even within professional tournaments. From a technical and tactical perspective, NA highlighted volley, smash and backhand strokes as the most common strokes among elite players [5,6,16,17]. However, the stroke distribution varies with age and gender, showing more strokes and lobs per rally in under-18 compared to younger male players (i.e., under-16) and vice-versa in under-16 female players [18]. Volleys, smashes and the low number of wall shots (e.g., side and back wall) represent an effective strategy in gaining an advantage when comparing shot effectiveness between winning and losing performances in elite players [19,20]. Indeed, these types of strokes may be advantageous, as they are executed in response to the opponent’s errors (e.g., shorter lobs to the opponent’s playing position) [6,21]. In fact, the smash after a lob is the most effective action to solve the point, although it is highly probable to end with an error [7]. From the defensive perspective, responding to smashes using an aggressive backwall defense could represent an effective and surprising counter-offensive strategy [7]. Moreover, smashes determining ball out could represent an effective strategy in scoring points for men more than for women, probably due to different strength levels between the genders [6].

Despite this important and growing information about performances in elite padel representing useful references for elite coaches and athletes, little is known about sub-elite or domestic competitions across Europe. From the match analysis perspective, which is the primary area of scientific interest from 2013 to date (i.e., 38 papers out of 72 reviewed), most (25 out of 38) focused on elite performance [10]. As a consequence, specific analyses for sub-elite competitions are needed. Performance is well known as level-dependent in padel [14,15], as in other sports [22,23], so specific research is needed to enhance scientific knowledge for athletes and technical staff. Therefore, this study aimed to describe sub-elite level padel competitions (i.e., the Italian second division “Serie B”) through technical, tactical and time-motion key performance indicators.

## 2. Materials and Methods

### 2.1. Design and Instruments

A descriptive-comparative study analyzed 4287 shots of 12 teams within 6 outdoor matches valid for the Italian Serie B male national league. Data were recorded from 10:00 15:00 local time (UTC+2) over a period of 2 days through a video camera (GoPro, Hero 4 Silver, GoPro Inc., San Mateo, CA, USA). The camera was located longitudinally with reference to the court, at a height of 4 m. The matches were recorded for their duration, and videos were saved in mp4 format. The software Longomatch Open Source version 1.3.2 installed on a MacBook Pro 15′′ (Apple, Cupertino, CA, USA) was utilized to analyze the matches. The local institutional review board approved this study (ID. 25831), and an informed consent form was obtained from the participants regarding the use of the video recordings for scientific purposes.

### 2.2. Methodology

According to previous research [10], 61 key performance indicators (KPIs)—9 for time-motion analysis (TMA) and 52 for notational analysis (NA)—were analyzed. They were clustered within five areas [time-motion analysis (TMA), shots characteristics (SC), errors (E), serve (S) and points won (PW)], and described as follows in Table 1. Since the measures in this study were mainly based on human perceptions, their reliability and objectivity represented an issue [24]. Therefore, a single match analyst (with more than three years of specific experience in notational analysis) analyzed all the matches to avoid any inter-observer variability. However, the match analyst and an expert padel coach examined an entire match randomly selected to assess reliability, reporting high inter-observer reliability for all KPIs (ICC range = 0.95 to 0.97). Finally, the intra-observer agreement was assessed by the match analyst, who analyzed 3 sets randomly chosen twice with an interval of 14 days, reporting a high intra-observer test-retest reliability (Intraclass Correlations, ICC = 0.99).

### 2.3. Data Analysis

Descriptive statistics was applied for the 11 TMA variables, and the data are presented as mean ± standard deviation. Due to the violation of normality, a non-parametric statistic was applied for the 48 NA variables. In particular, the Kruskall-Wallis test with Dunn’s post hoc was applied to analyze (i) the origin, (ii) the type of shots, (iii) winners, (iv) errors, and (v) the type of error in the serve. Differences among the type of the serve (forehand or backhand) and the biomechanics (overhead or under the head) of shots, errors and winners were investigated through the Mann-Whitney test. The significance level has been set at *p* < 0.05, and the effect size (ES) was calculated and interpreted accordingly: 0.2 to <0.6, small; 0.6 to <1.2, medium; 1.2 to <2.0, large; 2.0 to <4.0, very large; and ≥4.0, extremely large [25]. ICC was computed to determine intra- and inter-observer agreement.

## 3. Results

### 3.1. Time-Motion Analysis—(TMA)

The average match duration was 53.7 ± 1 min, divided into 16.8 ± 4 min effective playing time and 36.9 ± 11 min resting time. The effective playing time corresponded to 31.3% of the total match duration. The longest rally lasted 29.4 s. The average rally duration was 6.7 ± 1 s, while the recovery periods between rallies lasted 14.8 ± 2 s. On average, 42.6 ± 2.1 shots were played per minute of match. The WRR was 1:3.4 ± 0.8.

### 3.2. Notational Analysis—(NA)

Data in Table 2 reports all the significant differences, while shot characteristics (SC) are presented in Figure 1. On average, 4.7 ± 0.7 shots per rally were performed, while the average number of maximum shots played in a rally was 19.7 ± 4.

## 4. Discussion

The present study described Italian sub-elite level padel competitions (i.e., national second division “Serie B”) through technical, tactical and TMA key performance indicators. The results of this descriptive study represent a reference point for practitioners and coaches concerning the sub-elite padel performance model. Moreover, these results allow sports scientists to compare sub-elite to elite performance and highlight specific features of the sub-elite male matches in padel.

From the TMA perspective, sub-elite padel can be considered an intermittent sport as the elite one [17,26]. In fact, actions (i.e., shots) occurred frequently (e.g., 43 shots/minute) during short rallies (e.g., 6.7 s) at low density (e.g., WRR = 1:3.4). The effective playing time for elite players was heterogeneous, ranging from 35 to 46% [5,10]. However, the effective playing time reported in this study for sub-elite players (i.e., 31%) is lower than for elite. Indeed, performance level tends to decrease from the elite to the sub-elite level in terms of duration of the rallies, the number of shots and rate of play (i.e., strokes per minute) [15]. On the contrary, time-motion data in this study is in line with previous studies in elite padel, in terms of strokes per minute (~43) [15] and total match duration (~50 min) [5], but not for the WRR. Even though WRR varies between studies (i.e., 0.4 to 0.9) [13,27], the value reported in this study (i.e., 0.3) is significantly lower. On the one hand, these dissimilarities may be due to the normal fluctuation of the TMA variables within an open skill sport such as padel, while on the other hand, it could be due to a lack of consistency in data analysis. For instance, in this study, WRR was analyzed by averaging the ratio of the active (i.e., rally duration) and the subsequent recovery intervals, while other authors compared the total playing time with the total rest time [13].

From a technical and tactical perspective, shots were characterized by three important features: they were forehand volleys performed underhead. These evidences differ slightly from those reported for the elite level [28]. In fact, elite players perform more smashes and backhand strokes than sub-elite level [6,10,17,18], and they are even more efficient at volley shots [15]. Specifically, an offensive strategy is based on volleys shots to gain the net (i.e., to advance in a strategic position near the net) and smashes to score. On the other hand, the defensive strategy is based on sending the opponents to the backcourt and using the balls bouncing on the walls in the baseline [17]. In terms of efficacy, volleys and smashes were more likely to end with a point than the ground or wall shots, but they also led to errors. In fact, in this study, sub-elite winning pairs performed fewer volleys than the losing side (i.e., 69.1 vs. 80.9%) and fewer errors on volleys (i.e., 11 vs. 14.6%). On the contrary, elite winning players are more efficient at volley shots, and perform a significantly higher percentage of smashes and volleys and a lower number of ground and walls shots than the losing players [3,19]. This phenomenon may be explained by the different levels of players’ skills. Indeed, sub-elite players are generally less skilled, and could be more inclined to use less challenging shots (i.e., ground) rather than volleys or smashes [9]. In addition, less skill level and lower experience in high-level competitions may also influence the kinematics of the smash shots. Sub-elite players were reported to perform smashes at lower velocities than elite players, especially when affected by the opposition, to maximize the velocity-precision tradeoff [29]. Based on this evidence, one could speculate that greater smash and ball of the court errors in this study originated from shots executed at the high velocity at the expense of precision.

In this study, serves accounted for 21.3% of all shots, similarly to other sub-elite level performances [15]. One serve out of five (i.e., 22.6%) ended by error (vs. 8.8% for the elite level) [30], and almost half were net errors (i.e., 47.4%). In contrast to the elite level [30], a higher percentage of second serves occurred in this sample. In fact, sub-elite players struggled to perform successful first serves (i.e., 78.8%) compared to elite ones (i.e., 92.9). On the one hand, this evidence highlights the poor accuracy in executing the serve task, as well as an opportunity for the sub-elite players to gain an advantage by improving their serve skills and increasing their first serve efficacy. From the total errors that occurred in a match perspective, serve errors accounted for 30.6%. Special focus should be put on the serve and the serve return, since it was suggested that their quality could influence the rally outcome and duration, especially for sub-elite players [15,31]. According to Ramón-Llin et al., a good or bad serve could anticipate the end of the rally and lead to shorter rally durations [15]. Similarly, in professional padel, the beginning of each point is very important and decisive for increasing the chances of winning the point [30]. In particular, the beginning of each point consists of both serve and serve return. Hence, the serve effectiveness is directly related to the opponent’s serve return skills [32]. However, serves did not distinguish between winners and losers in this study. Conversely, fewer errors during serve return shots were advantageous for the winning pairs in this study (i.e., 9.3 vs. 18.7%). In general, serve return error percentage in padel is lower than in other sports such as tennis, due to the lower serve power [32]. From the regulation perspective, an underhand shot from a bouncing ball is mandatory for the padel’s serve, so it is relatively easy to play. In fact, in this study, aces (2%) and double faults (1.6%) were even lower than in elite tennis tournaments (e.g., Wimbledon) [33]. Therefore, sub-elite players should focus on the serve return technical and tactical skills to prevent the server from winning the rally quickly [30]. In fact, according to Ramón-Llin et al., the server has a significant advantage in padel, especially during short rallies up to 6 to 8 shots [31].

Finally, error analysis for the winning rallies (see Figure 1d) highlights that unforced errors caused 65% of the points scored. The reason for this phenomenon may be due to the poor technical and tactical skill level of the sub-elite players. This scenario is consistent with the differences between winning and losing players presented in this study. From the quantitative perspective, winning players tend to perform less challenging actions (e.g., volleys), take fewer risks and let the opponent make the error, while from the qualitative perspective, winning players are also more effective (i.e., make fewer errors) when performing (e.g., volleys, smashes, serve returns) the shots.

Nevertheless, this study presents some limitations. First, we did not provide evidence on the kinematic match demands such as distance covered, velocities, accelerations and decelerations, change of directions and type of displacement (e.g., standing, walking, running, sprinting) to describe the physical load. We only assumed TMA components such as effective playing time (i.e., volume) and WRR (i.e., density) to provide information on physical load. Secondly, we did not provide insights about scoring strategies and players’ location (i.e., baseline or net) according to winning and losing matches. Therefore, further research on the kinematic match demands through GPS or Video tracking technology, as well as on the players’ offensive and defensive strategies concerning the match outcome (i.e., winning/losing), is needed in sub-elite competitions to compare technical and tactical patterns with elite ones [34]. Finally, information about the participants (i.e., average age, number of official matches played per year, national league players success ranking) is not reported. Thus, caution is necessary when interpreting these data.

## 5. Conclusions

This study presented new contributions to performance indicators in sub-elite padel competitions. Data suggested that the sub-elite matches showed a lower density and less effective playing time than elite ones. The results indicated that when the points were scored, the more challenging shots were used (e.g., volleys, smashes). However, more challenging shots also resulted in more unforced errors. In fact, when analyzing outcomes from the winning and losing players perspective, the winning sub-elite players generally performed easier shots (i.e., ground) than volleys or smashes, adopted a more conservative playing style, and let the opponents to commit unforced errors. Data also suggested that serves, especially the serve returns, may be key factors to train. Indeed, training technical and tactical skills to prevent the server from winning the rally quickly may be a pivotal strategy in sub-elite padel. This information may contribute to the existing knowledge in padel for setting benchmarks and adapting training plans specifically for sub-elite competitions. However, since technical and tactical performance is closely linked to the physical one (e.g., strength and conditioning) [17], future studies should focus on analyzing the sub-elite level in terms of physical fitness, strength and conditioning training, and time-motion analysis.

## Figures and Tables

**Figure 1 ijerph-19-08386-f001:**
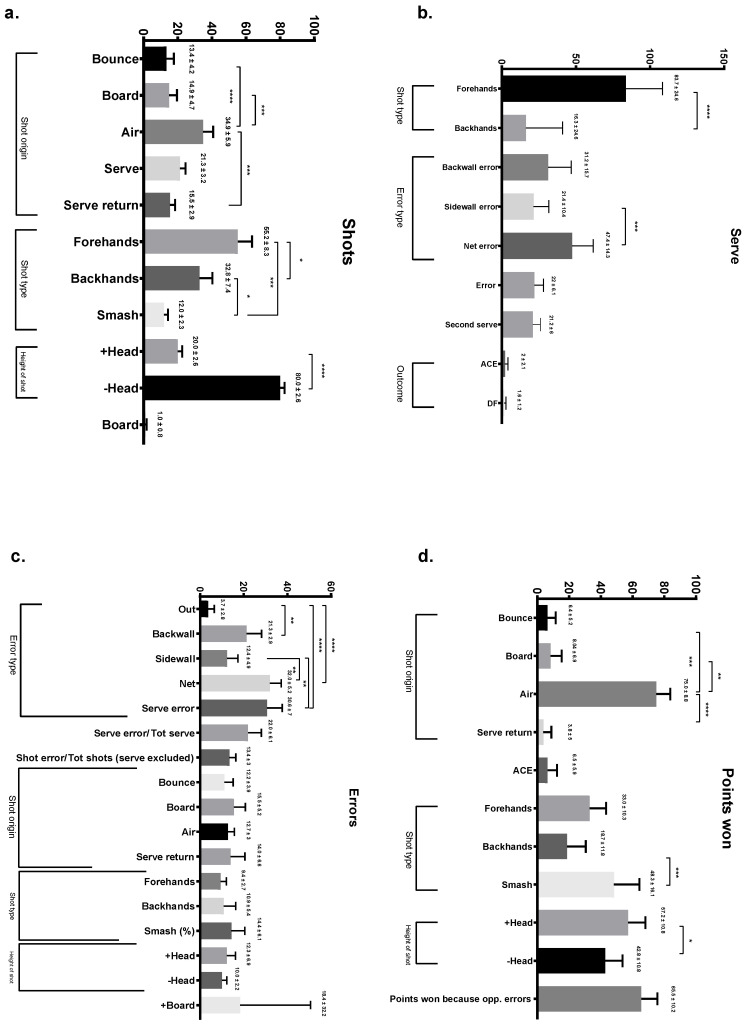
Characteristics of shots (**a**), serve (**b**), errors (**c**) and points won (**d**). Data are expressed as mean percentage ± standard deviation. * *p* ≤ 0.05; ** *p* ≤ 0.01; *** *p* ≤ 0.001; **** *p* ≤ 0.0001.

**Table 1 ijerph-19-08386-t001:** Description of the KPIs according to the 5 areas of investigation: time-motion analysis (TMA), shots characteristics (SC), errors (E), serve (S) and points won (PW).

Area	Performance Indicators	Description
**Time-motion Analysis (TMA)**	Total playing time	Effective playing time + total recovery time (min)
	Effective playing time	Effective playing time from the serve to the point scored (min)
	Total recovery time	Total recovery time from the point scored to the next serve (min)
	Average rally duration	Average duration of the time intervals from the serve to the point(s) scored
	Maximum rally duration	Maximum duration of the longest time interval from the serve to the point(s) scored
	Average recovery between rallies	Average duration of the time intervals from the point scored to the next serve(s)
	Maximum recovery between rallies	Maximum duration of the time interval from the point scored to the next serve(s)
	Work-to-rest ratio (WRR)	Average of the ratios between rally duration and the following recovery time
	Shots per minute	Ratio between the total number of shots performed by both the teams and the effective playing time
**Shot characteristics (SC)**		
*Origin*	Bounce	Shot performed after a bounce of the ball
	Board	Shot performed after the ball touches against the sidewall and/or backwall
	Air (volley)	Shot performed without any previous bounces of the ball
	Serve	Shot to start a point
	Serve return	Shot performed after the serve of the counterpart
*Type of shot*	Forehand	Shot performed with the forehand
	Backhand	Shot performed with the backhand
	Smash	Shot performed with the smash (including flat, topspin, tray)
*Height of shot*	Overhead	Shot in which the ball is at the level of the ear or above it when impacted by the racket
	Underhead	Shot in which the ball is at the level of the ear or below it when impacted by the racket
*Board*	Yes	Shot arrived in the counterpart’s court after touching the board in their own court
	No	Shot arrived in the counterpart’s court without touching the board in their own court
Average number of shots per rally Maximum number of shots per rally		Average number of shots performed by both teams each rally Maximum number of shots performed by both teams in one rally
*Total*		Total number of shots performed
**Errors (E)**		
*Type of error*	Out	Shot ending with a point for the opponent because of the ball sent out of the court
	Length	Shot ending with a point for the opponent because the ball was sent directly on the backwall
	Width	Shot ending with a point for the opponent because the ball was sent directly on the sidewall or side fences
	Net	Shot ending with a point for the opponent because the ball was blocked by the net
	Errors at the serve	Total number of wrong serves
*Origin*	Bounce	Shot ending with a point for the opponent performed after a bounce of the ball
	Board	Shot ending with a point for the opponent performed after the ball touched against the sidewall or backwall
	Air (volley)	Shot ending with a point for the opponent performed before any bounces of the ball
	Serve return	Shot ending with a point for the opponent performed after the serve of the counterpart
*Type of shot*	Forehand	Shot ending with a point for the opponent performed with the forehand
	Backhand	Shot ending with a point for the opponent performed with the backhand
	Smash	Shot ending with a point for the opponent performed with the smash (including flat, topspin, tray)
*Height of shot*	Overhead	Shot ending with a point for the opponent in which the ball is at the level of the ear or above it when impacted by the racket
	Underhead	Shot ending with a point for the opponent in which the ball is below the level of the ear when impacted by the racket
*Board*	Yes	Shot ending with a point for the opponent sent in the counterpart’s court after having touched one board in the own court
	No	Shot ending with a point for the opponent sent in the counterpart’s court without having touched one board in the own court
*Total errors/total shots*		Ratio between the total number of errors and the total number of shots performed
**Serve (S)**		
*Type of shot*	Forehand	Serve performed with the forehand
	Backhand	Serve performed with the backhand
*Serve number*	First	First shot starting the rally (according to the Rule n.6 of the International Padel Federation) [4]
	Second	Second shot if the first was not valid
*Type of error*	Out	Serve not valid because of the ball sent out of the court
	Length	Serve not valid because of the ball touching directly the backwall
	Width	Serve not valid because of the ball touching directly the sidewall or touching the side fence after the bounce
	Net	Serve not valid because of the ball blocked by the net
*Total serves/error during serve*		Ratio between the total number of errors during serve and the total number of serves performed by a player
**Points won (PW)**		
*Origin*	Bounce	Shot performed after a bounce of the ball
	Board	Shot performed on a ball returning from a touch against the sidewall or backwall
	Air (volley)	Shot performed before any bounces of the ball
	Ace	Serve to allow for scoring the point before the counterpart touches the ball
	Serve return	Shot performed after the serve of the counterpart
*Type of shot*	Forehand	Shot performed with the forehand that ends with a point scored
	Backhand	Shot performed with the backhand that ends with a point scored
	Smash	Shot performed with the smash (including flat, topspin, tray) that ends with a point scored
*Height of shot*	Overhead	Shot ending with a point scored in which the ball is at the level of the ear or above it when impacted by the racket
	Underhead	Shot ending with a point scored in which the ball is below the level of the ear when impacted by the racket
*Board*	Yes	Shot ending with a point scored that arrives in the counterpart’s court after having touched one board in the own court
	No	Shot ending with a point scored that arrives in the counterpart’s court without having touched one board in the own court
** *Points won by means of the opponents’ mistakes* **		Total number of points won following an opponent’s mistake (i.e., opponents’ unforced errors), in a technical and tactical situation where the opponent was not constricted to respond after a high effective shot (e.g., high-speed ball, very close to the wall ball).
** *Total* **		Total number of points won by means of winners + total number of points won by means of opponents’ mistakes

**Table 2 ijerph-19-08386-t002:** Results of the inferential statistics applied to the notational analysis KPIs.

Areas	Performance Indicators	*p*	ES	diff %
Shot characteristics				
*Origin*	volley vs. bounce	****	4.4	34.9 ± 5.9 vs. 13.4 ± 4.2
	volley vs. service return	***	4.4	34.9 ± 5.9 vs. 15.5 ± 2.9
	volley vs. board	***	3.9	34.9 ± 5.9 vs. 14.9 ± 4.7
*Type of shot*	forehand vs. backhand	*	3.0	55.2 ± 8.3 vs. 32.8 ± 7.4
	forehand vs. smash	***	7.4	55.2 ± 8.3 vs. 12 ± 2.3
	backhand vs. smash	*	4.0	32.8 ± 7.4 vs. 12 ± 2.3
*Height of shot*	under vs. over the head	****	24.6	80.0 ± 2.6 vs. 20.0 ± 2.6
**Errors**				
*Type of error*	serve vs. ball out of the court	****	5.3	30.6 ± 7 vs. 3.7 ± 2.8
	serve vs. width	**	3.2	30.6 ± 7 vs. 12.4 ± 4.9
	net vs. out	****	7.1	32.0 ± 5.2 vs. 3.7 ± 2.8
	net vs. width	**	4.1	32.0 ± 5.2 vs. 12.4 ± 4.9
	longboard vs. ball out of the court	**	3.5	21.3 ± 6.9 vs. 3.7 ± 2.8
**Serve**				
*Type of shot*	forehand vs. backhand	****	2.9	83.7 ± 24.6 vs. 16.3 ± 24.6
*Type of error*	net vs. sidewall	***	2.2	47.4 ± 14.3 vs. 21.4 ± 10.4
**Points won**				
Origin	volley vs. bounce	***	10.0	75.0 ± 8.8 vs. 6.4 ± 5.2
	volley vs. service return	****	10.5	75.0 ± 8.8 vs. 3.8 ± 5
	volley vs. board	**	8.8	75.0 ± 8.8 vs. 8.4 ± 6.9
*Type of shot*	smash vs. backhand	***	2.2	48.3 ± 16.1 vs. 18.7 ± 11.8
*Height of shot*	overhead vs. under the head	*	1.4	57.2 ± 10.8 vs. 42.8 ± 10.8
**Winning (W)/Loosing (L)**				
Points won	volley: (W) vs. (L)	*	2.0	69.1 ± 6.2 vs. 80.9 ± 6.7
Errors	ball out of the court: (W) vs. (L)	*	1.6	2.1 ± 2.5 vs. 5.4 ± 2.0
	errors/shots ratio: (W) vs. (L)	**	2.5	11.5 ± 1.3 vs. 15.45 ± 2.3
	volley: (W) vs. (L)	*	1.7	11.0 ± 1.8 vs. 14.6 ± 2.9
	serve return: (W) vs. (L)	**	2.2	9.3 ± 3.3 vs. 18.7 ± 5.7
	backhand: (W) vs. (L)	*	1.3	8.0 ± 2.6 vs. 13.7 ± 6.2
	smash: (W) vs. (L)	**	2.2	10.0 ± 2.8 vs. 18.7 ± 5.4
	overhead shots: (W) vs. (L)	*	1.5	10.0 ± 2.7 vs. 14.6 ± 3.8

Notes: * = *p* < 0.05, ** = *p* < 0.01, *** = *p* < 0.001, **** = *p* < 0.0001.

## Data Availability

Not applicable.

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
