# Peer review of "Padel Match Analysis: Notational and Time-Motion Analysis during Official Italian Sub-Elite Competitions"

_ijerph, 2022, doi:10.3390/ijerph19148386_

Round 1

Reviewer 1 Report

General comments

The authors should be more precise with the specific vocabulary. Please use the word "match" instead of the word "game" to refer to complete matches, use the word "wall" instead of "board" to refer to the walls of the court. Replace the terms "sideboard" and "longboard" with "sidewall" and "backwall".

Introduction

The reference provided in the statement "the stroke distribution varies with age and gender, showing more strokes and lobs per rally in under-18 compared to younger male players (i.e., under-16) and vice-versa in under-16 74 female players" is not correct. The article referred to does not analyse young players.

Methodology

The classification used to group the performance indicators is not adequate. Also, the definition and number of performance indicators should be improved.

·      Shot characteristics

o   Origin: Hits executed after bouncing off two walls are not included.

o   Type of shot: The indicators included in this category are insufficient. For example, they do not include the tray shot.

o   Biomechanics: The title of this category is not appropriate as biomechanics refers to many other aspects besides whether the ball is hit over or under the head. Furthermore, the authors should justify the need to include this category. If all other categories were adequate, this one would not be necessary.

o   Total: Why are total strokes included in the "Shot characteristics" category and strokes per point in "TMA"?

·      Errors

o   Type of error: Again, the indicators included in this category are insufficient. For example, there is no indication of how errors were recorded when the ball was sent directly to the side fence. A distinction should also be made between first and second service errors, in addition to some aspects cited in the previous area that are also applicable to this one.

·      Serve 

o   Type of shot: Should include whether it is first or second service. 

o   Type of error: Services which, after the bounce, hit the side fence are not included.

·      Points won

o   Origin: Direct points with the service "aces" are specified but not direct points with the rest of the strokes "Winners".

o   Points won by means...: The definition is not precise "in a situation that should be fully controlled by the player". Furthermore, it is not specified what happens in points that end with a forced error.

The authors should have relied on articles that have previously defined categories and indicators more precisely.

The discussion and results sections should be completely rewritten (with the exception of the information on MAT) once the indicators and categories have been corrected.

Reviewer 2 Report

General comments

Padel match analysis: notational and time-motion analysis during official Italian sub-elite competitions.

The authors have done a great job of providing an informative and meaningful addition to the current study field.

However, there are several changes that the authors are encouraged to revise to elevate the overall contribution of the paper to this research field.

Introduction

The authors should add the relevant references about the “Notational Variables and time motion on tennis and padel players” these studies will allow improving the manuscript.

The authors should add these articles

Line 71      winning and losing players according to players

Escudero-Tena, A., Sánchez-Alcaraz, B. J., García-Rubio, J., & Ibáñez, S. J. (2021). Analysis of Game Performance Indicators during 2015–2019 World Padel Tour Seasons and Their Influence on Match Outcome. International journal of environmental research and public health, 18(9), 4904.

Materials and methods

How many days were the matches played?

what time of day was it played?

Participants

Please provide more information about the participants

for instance:    

                              Participants average age

                              Number of official matches played per year

                              Italian Serie B male national league players success ranking

Discussion

Line 199    time-motion analysis, shots characteristics.

Sánchez-Alcaraz, B. J., Martínez-Gallego, R., Ramón-Llin Mas, J., Crespo, M., Muñoz, D., López Martínez, J. M., & Sánchez-Pay, A. (2022). Professional padel tennis: Characteristics and effectiveness of the shots played to the fence. International Journal of Sports Science & Coaching, https://doi.org/10.1177/17479541221093765

Line 215      technical and tactical perspective

Ramon-Llin, J., Sanchez-Alcaraz, B. J., Sanchez-Pay, A., Guzman, J. F., Vuckovic, G., & Martínez-Gallego, R. (2021). Exploring offensive players’ collective movements and positioning dynamics in high-performance padel matches using tracking technology. International Journal of Performance Analysis in Sport, 21(6), 1029-1040.

References

- Reference writing in the entire article should be checked

Line 303   (add date of access and URL if applicable)

Line 345

Reviewer 3 Report

It is a good study that contributes to knowledge, however, it is necessary to correct some details such as:

It is suggested to start by indicating the design and not start with the sample. An example would be: This descriptive-comparative study analyzed a sample of 4,287 shots from 12 teams cross-sectionally...

The approval number or code is missing, which allows verifying the ethical commitment of the study

It is required to incorporate a paragraph or section where the instruments are described. It should mention issues such as relevance and validity.

It is required to incorporate a paragraph or section where the instruments are described. It should mention issues such as relevance and validity.

What are the strengths and limitations of the study?

I propose to eliminate the last suggestion since nothing indicates anthropometric variables, body composition, physical condition, etc. during the study.

Reviewer 4 Report

The study is devoted to the analysis of the technical armament of non-elite level padel players. A fairly modern video recording method was used for the study, followed by an analysis of the performance of technical actions and errors by athletes. The sample of the study is large enough for analysis and drawing up a conclusion. At the same time, there is no novelty in the results obtained, as evidenced by the conclusion of the authors of the study, which is also very sparingly reflected in the abstract of the work. In fact, the work has the character of a descriptive presentation of materials, despite the existing discussion, the question of the novelty of the materials received, according to the reviewer, has not been resolved by the authors.

Round 2

Reviewer 1 Report

Thank you for considering the proposals made in the previous review.

Congratulations on the article

Author Response

Thank you for the congratulations and contribution in improving the quality of the manuscript

Reviewer 4 Report

The study is devoted, in general, to a fairly new (non–Olympic) sport that is actively developing in the Rada of countries - padel. It should be noted that the authors have made quite significant improvements to the text of the article in general. At the same time, the work in question is undoubtedly of interest from the standpoint of the methodology of this sport and does not fit at all into the stated goals of the journal – assessing the impact of external environmental factors on the human body and public Health.  This work may have a certain methodological interest from the position of the coach's work, and is devoted to the analysis of errors in the sports discipline in question and refers to the methodology and methodology of training athletes in this sport, and not the influence of external environmental factors on the human body. With the conclusions of the work, the authors present the new performance indicators described by them in the sub-elite padel competition, which has nothing to do with the tasks of the journal according to the reviewer. The additional indicators described by the authors related to errors and insufficiently qualified athletes in general have a narrowly focused interest and are quite obvious and understandable.

Author Response

Authors: Although we respect the reviewer’s point of view, we do believe that the data, the analysis, the rationale, and the story of the present study are more than adequate in terms of scientific rigor and content and is fully pertinent for this journal. In fact, similar studies were just published in the International Journal of Environmental Research and Public Health (e.g., Analysis of Game Performance Indicators during 2015–2019 World Padel Tour Seasons and Their Influence on Match Outcome; Performance Outcome Measures in Padel: A Scoping Review; Match Analysis, Physical Training, Risk of Injury and Rehabilitation in Padel: Overview of the Literature; Analysis of the Actions of Net Zone Approach in Padel: Validation of the NAPOA Instrument).  At the same time, the journal provided and still provides different Special Issues about sports performance, sports methodology, and methodology of training (e.g., “Sport Modalities”, “Performance and Health; Impact of Racket Sports”; “Fitness, Physical Education, Physiological Responses and Health Promotion”; and the current “Team Sports Implications for Training Load, Performance Analysis and Health”).
We admit that padel match analysis has been already treated in literature, but very little research was developed on sub-elite padel players, and no study has been provided on Italian sub-elite competition level, making this study original. In addition, we lowly believe that this study substantially improved its quality thanks to the reviewers’ suggestions provided in the process and can be considered as fully suitable to this Special Issue within the International Journal of Environmental Research and Public Health journal.